# MODEL-BASED NAVIGATION IN ENVIRONMENTS WITH NOVEL LAYOUTS USING ABSTRACT 2-D MAPS

## ABSTRACT

Efficiently training agents with planning capabilities has long been one of the major challenges in decision-making. In this work, we focus on *zero-shot navigation ability* on a given abstract 2-D occupancy map, like human navigation by reading a paper map, by treating it as an *image*. To learn this ability, we need to efficiently train an agent on environments with a small proportion of training maps and share knowledge effectively across the environments. We hypothesize that *model-based* navigation can better adapt agent's behaviors to a task, since it disentangles the variations in map layout and goal location and enables longer-term planning ability on novel locations compared to reactive policies. We propose to learn a *hypermodel* that can understand patterns from a limited number of abstract maps and goal locations, to *maximize alignment* between the hypermodel predictions and real trajectories to extract information from multi-task off-policy experiences, and to construct denser feedback for planners by $n$-step goal relabelling. We train our approach on DeepMind Lab environments with layouts from different maps, and demonstrate superior performance on zero-shot transfer to novel maps and goals.

## 1 INTRODUCTION

If we provide a rough solution of a problem to an agent, can the agent learn to follow the solution effectively? In this paper, we study this question within the context of maze navigation, where an agent is situated within a maze whose layout has never been seen before, and the agent is expected to navigate to a goal without first training on or even exploring this novel maze. This task may appear impossible without further guidance, but we will provide the agent with additional information: an abstract 2-D occupancy map illustrating the rough layout of the environment, as well as indicators of its start and goal locations ("task context" in Figure 1). This is akin to a tourist attempting to find a landmark in a new city: without any further help, this would be very challenging; but when equipped with a 2-D map with a "you are here" symbol and an indicator of the landmark, the tourist can easily plan a path to reach the landmark without needing to explore or train excessively.

Navigation is a fundamental capability of all embodied agents, both artificial and natural, and therefore has been studied under many settings. In our case, we are most concerned with *zero-shot navigation* in novel environments, where the agent cannot perform further training or even exploration of the new environment; all that is needed to accomplish the task is technically provided by the abstract 2-D map. This differs from the vast set of approaches based on simultaneous localization and mapping (SLAM) typically used in robot navigation (Thrun et al., 2005), where the agent can explore and build an accurate occupancy map of the environment prior to navigation. Recently, navigation approaches based on deep reinforcement learning (RL) approaches have also emerged, although they often require extensive training in the same environment (Mirowski et al., 2017; 2018). Some deep RL approaches are even capable of navigating novel environments with new layouts without further training; however, these approaches typically learn the strategy of efficiently exploring the new environment to understand the layout and find the goal, then exploiting that knowledge for the remainder of the episode to repeatedly reach that goal quickly (Jaderberg et al., 2017). In contrast, since the solution is essentially provided to the agent via the abstract 2-D map, we require a more stringent version of zero-shot navigation, where it should not explore the new environment; instead, we expect the agent to produce a near-optimal path in its *first* (and only) approach to the goal.

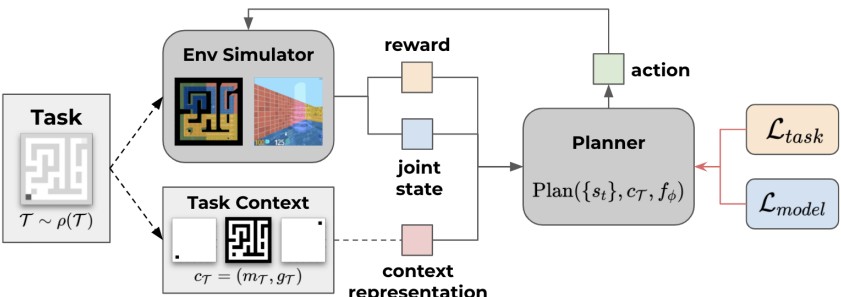

Figure 1: In each training episode, we randomly select a task $\mathcal{T}$ to initialize environment simulation and feed the corresponding task context $c_\mathcal{T}$ to the agent. We use a joint state space $o \in \mathbb{R}^{12}$ as input to the agent, consisting of position $\mathbb{R}^3$, orientation $\mathbb{R}^3$, and translational and rotational velocity $\mathbb{R}^6$. Each cell on the abstract map corresponds to 100 units in the agent world.

Although the solution is technically accessible via the abstract 2-D map, some challenges remain to use it effectively. First, although we assume that the layout in the 2-D map is accurate, the map does not correspond to the state space of the agent in the environment, so the agent must learn the correspondence between its state and locations in the 2-D map. Second, actions in the 2-D map also cannot be directly mapped into the agent's action space; moving betweend adjacent "cells" in the 2-D map requires a sequence of many actions, specified in terms of the agent's translational and rotational velocities. Hence, one cannot simply perform graph search on the 2-D map, then execute the abstract solution directly on the agent. Instead, we propose approaches that learn to use the provided abstract 2-D map via end-to-end learning.

Concretely, we propose two approaches for navigation using abstract 2-D maps:

- **MMN** (Map-conditioned Multi-task Navigator): A model-based approach that learns a *hyper-model* (Ha et al., 2016), which uses the provided 2-D map to produce a parameterized latent-space transition function $f_\phi$ for that map. This transition function $f_\phi$ is jointly trained with Monte-Carlo tree search (MCTS) to plan (in latent space) to reach the specified goal Schrittwieser et al. (2019).
- **MAH** (Map-conditioned Ape-X HER DQN): A model-free approach based on Ape-X Deep Q-Networks (DQN) (Horgan et al., 2018), a high-performing distributed variant of DQN, that takes in the provided 2-D map as additional input. Furthermore, we supplement it with our proposed $n$-step modification of hindsight experience replay (HER) (Andrychowicz et al., 2017).

In experiments performed in DeepMind Lab (Beattie et al., 2016), a 3-D maze simulation environments shown in Figure 1, we show that both approaches achieve effective zero-shot navigation in novel environment layouts, and the model-based **MMN** is better at long-distance navigation.

## 2 BACKGROUND

We consider a distribution of navigation tasks $\rho(\mathcal{T})$. Each task is different in two aspects: map layout and goal location. (1) *Abstract map*. The layout of each navigation task is specified by an abstract map. Specifically, an abstract map $m \in \mathbb{R}^{N \times N}$ is a 2-D occupancy grid, where cell with 1s (black) indicate walls and 0s (white) indicate nagivable spaces. A cell does not correspond to the agent's world, so the agent needs to learn to localize itself given an abstract 2-D map. We generate a set of maps and guarantee that any valid positions are reachable, i.e., there is only one connected component in a map. (2) *Goal position*. Given a map, we can then specify a pair of start and goal position. Both start and goal are represented as a "one-hot" occupancy grid $g \in \mathbb{R}^{2 \times N \times N}$ provided to the agent. For simplicity, we use $g$ to refer to both start and goal, and we denote the provided map and start-goal positions $c = (m, g)$ as the *task context*.

We formulate each navigation task as a goal-reaching *Markov decision process* (MDP), consisting of a tuple $\langle \mathcal{S}, \mathcal{A}, P, R_\mathcal{G}, \rho_0, \gamma \rangle$, where $\mathcal{S}$ is the state space, $\mathcal{A}$ is the action space, $P$ is the transition function $P : \mathcal{S} \times \mathcal{A} \to \Delta(S)$, $\rho_0 = \rho(s_0)$ is the initial state distribution, and $\gamma \in (0, 1]$ is the discount factor. We assume transitions are deterministic. For each task, the objective is to reach a

subset of state space $\mathcal{S}_\mathcal{G} \subset \mathcal{S}$ indicated by a reward function $R_\mathcal{G} : \mathcal{S} \times \mathcal{A} \to \mathbb{R}$. We denote a task as $\mathcal{T} = \langle P, R_\mathcal{G}, \rho_0 \rangle$, since a map and goal specify the dynamics and reward function of a MDP, respectively. In the episodic goal-reaching setting, the objective is typically not discounted ($\gamma = 1$) and the reward is $-1$ for all non-goal states, i.e., $R_\mathcal{G}(s, a) = -\mathbb{I}[s \neq g], g \in \mathcal{S}_\mathcal{G}$.

## 3 MAP-CONDITIONED PLANNING GIVEN ABSTRACT 2-D MAPS

To build a map-based navigation agent efficient in both training and transfer, there are several technical challenges. (1) A local change in map may introduce entirely different task structure, so we need the model and planner to adapt to the task context in a different way than conditioning on state, and not directly condition on the entire task context. (2) During training, we can only rely on a very small proportion of training tasks (e.g., 20 of $13 \times 13$ maps). This requires the agent to be efficient in terms of understanding each task's structure, i.e., the layout of training maps and goal locations. (3) Since reward is sparse and model learning and exploration are done simultaneously, we need to fully utilize the knowledge in the environment to train the model and planner, such as transition tuples and failure experiences. Corresponding to the challenges, we introduce the task-conditioned hypermodel first, followed by the *forward* pass of the planning computation by using the hypermodel in inference. We then detail the *backward* pass on the training target and optimization process.

### 3.1 TASK-CONDITIONED HYPERMODEL

We aim to build a model adaptive to given abstract 2-D maps for the navigation planner. In a single-task training schema, a naive solution is to separately learn a parameterized transition function $f_i(s, a)$ for different maps. However, we want to share knowledge between tasks in navigation domain, in which maps have some common computational patterns. For example, in Figure 2, moving right on center of the box on the left map shares some computation with the right one. This also applies to the larger scale of map area and also the reward prediction. When the agent is able to capture this type of computational pattern, it can better predict what will happen when transferring to a new task.

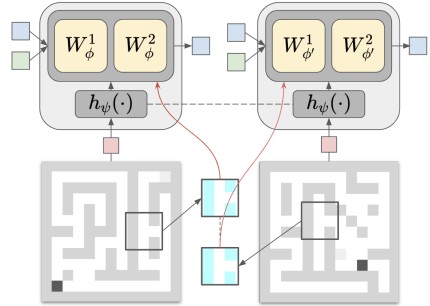

Figure 2: Applying a hypermodel $h_\psi$ on two maps. They may share local patterns at some scales that can be captured by the hypermodel.

We propose to build a *meta* network $h_\psi$, or *hypermodel*, to learn the "computation" of the transition model $f_\psi$ simultaneously for all maps with abstract 2-D maps as input. The transition model for task $\mathcal{T}$ (map-goal pair) is a function $f_i$ that maps current (latent) state and action to a next (latent) state. The set $\{f_i\}$ represents transition functions of all tasks belonging to a navigation schema (e.g., a certain size of map), and these tasks have similar structure. We parameterize a transition function $f_i$ as a neural network with its parameter vector $\phi_i$. We assume the set of transition networks have similar structure that can be characterized by a set of context variables $c = (m, g)$, i.e., the abstract 2-D map and goal.[1] This implies that parameter vectors $\phi_i$ live in a low-dimensional manifold. Thus, we define a mapping $h : \mathcal{C} \to \Phi$ that maps the context of a task to the parameter vector $\phi_i$ of its transition function $f_i$. We parameterize $h$ also as a network with parameter $\psi$:[2]

$$h_\psi : c \mapsto \phi, \quad f_\phi : s, a \mapsto s'. \tag{1}$$

This can be viewed as soft weight sharing between multiple tasks. It efficiently maps low-dimensional structure in the MDP, specified by the map, to computation of the transition model. It may also be viewed as a learned "dot-product" between task context $c_\mathcal{T}$ and state and action $s_t, a_t$ to predict the next state. The idea of predicting the weights of a main network using another *meta*-network is also known as HyperNetworks (Ha et al., 2016; von Oswald et al., 2019).

---

[1] Concretely, a task context $c \in \mathbb{R}^{4 \times N \times N}$ has four components: downsampled global occupancy map, cropped local occupancy map, and "one-hot" goal and start occupancy maps, where $N$ is downsampled size.

[2] We only predict weights of the transition model $f_\phi : \mathcal{S} \times \mathcal{A} \to \mathcal{S}$ which operates on a latent state space. The mapping from environment observations to latent states $e : \mathcal{O} \to \mathcal{S}$ is not predicted by a meta network. Since the latent space is low-dimensional, it is feasible to predict weight matrices of a transition network for it.

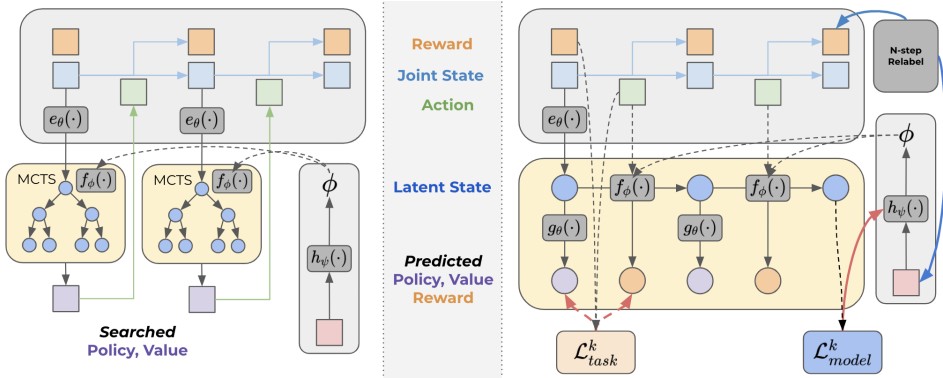

Figure 3: We use yellow or circles to indicate predicted states or other quantities, and grey or squares from actual interactions. (left) Applying MCTS with hypermodel to search for behavioral policy and value, and act with a sampled action. (right) Computing targets and backpropagating from loss. The blue line indicates $n$-step relabelling. We only illustrate backpropagation for one reward node for simplicity. The solid red line is to emphasize the gradient flow from the auxiliary model loss to the meta network's weight $\psi$. The dashed red line is the gradient from task loss.

## 3.2 PLANNING USING LEARNED HYPERMODEL

Equipped with a map-conditioned model, we use it to search actions according to the map layout and goal location $(a^1, ..., a^k) = \text{Plan}(\{s_i\}, c, f_\phi)$. We use Monte Carlo tree search (MCTS) (Silver et al., 2017; Schrittwieser et al., 2019) to search with learned hypermodel $f_\phi$. For multi-task, a special benefit is that model-based planning disentangles the navigation behavior in terms of goal level and map level, which correspond to reward function and transition dynamics, respectively. Intuitively, the navigation computation $\text{Plan}(\{s_i\}, c, f_\phi)$ is highly non-linear in $c = (m, g)$, and thus disentangling them leads to more effective learning and generalizing across tasks.

As shown in Figure 3, we first encode the observed joint state $o_t$ into a latent space $s_t$ with learned encoder $e_\theta(o_t)$, as the root node of the search tree. Given a latent state and a candidate action, the next state is predicted using hypermodel $f_\phi$. For each state (node), another network $g_\theta(s_t, a_t)$ predicts the policy $\pi_t$ and value function $v_t$ (Schrittwieser et al., 2019), which are used for aligning with searched quantities in a separate training process. After a number of simulations is done and the statistics are backpropagated to the root node, we sample an action from the searched policy. The trajectory and corresponding abstract map and goal $(c_\mathcal{T}, \{s_t, a_t, r_t, s_{t+1}\}_t)$ are saved to a centralized replay buffer. In the training process, these quantities are predicted on latent states and aligned to the saved quantities from actual experiences.

At **zero-shot evaluation** time, given an unseen abstract map, we apply planning with trained hypermodel: (1) given a map and goal $c_\mathcal{T} = (m_\mathcal{T}, g_\mathcal{T})$, at the beginning of the episode, compute the hypermodel weights $\phi = h(c; \psi)$ by applying the meta-network on the task context $c_\mathcal{T}$, (2) start MCTS simulations using the hypermodel $f(s, a; \phi)$ for latent state predictions, (3) get an action and transit to next state, and go to step (2) and repeat. Moreover, if we assume to have a *landmark oracle* on given maps, we can perform **hierarchical navigation** by generating a sequence of local subgoals $\{(m, g_i)\}_{i=1}^n$, and plan to sequentially achieve each intermediate landmark.

## 3.3 CONSTRUCTING LEARNING TARGETS WITH $n$-STEP GOAL RELABELLING

Jointly training a planner with learned model can suffer from lack of reward signal, especially when the model training entirely relies on reward from *multiple tasks*, which is common in model-based agents based on *value gradients* (Schrittwieser et al., 2019; Oh et al., 2017). Motivated by this, we introduce a straightforward strategy to enhance the reward signal by implicitly defining a learning curriculum, named *n-step hindsight goal relabelling*. This generalizes the single-step version of *Hindsight Experience Replay* (HER) (Andrychowicz et al., 2017) by extending it to $n$-step return.

We adopt a multi-step strategy motivated by single-step HER, by relabelling failed goals to randomly sampled *future* states (visited area) from the trajectory, and associating states with the relabelled $n$-

step return. Concretely, the *task-conditioned* bootstrapped $n$-step return is

$$G_t^{\mathcal{T}} \doteq r_{t+1} + \gamma r_{t+2} + \gamma^2 r_{t+3} + \cdots + \gamma^n v_n^{\mathcal{T}}, \quad v_n^{\mathcal{T}}, \pi_n^{\mathcal{T}} = g_\theta(s_t, c_{\mathcal{T}}) \qquad (2)$$

where $v_n^{\mathcal{T}}$ is the the state-value function bootstrapping $n$ steps into the future from the search value and conditioned on task context $c_{\mathcal{T}}$. Empirically, this strategy significantly increases the efficiency of our multi-task training by providing smoothing gradients when sampling a *mini-batch* of $n$-step targets from successful or failed tasks. Additional details are provided in the appendix.

### 3.4 JOINT OPTIMIZATION

Our training target has two components. The first component is based on value gradients (Schrittwieser et al., 2019; Oh et al., 2017), where the gradient from value predictions on predicted experiences is backpropagated from the aforementioned (relabelled) $n$-step targets. However, the value-gradient-based method is designed for single-task RL, which can be sample inefficient in training on different map layouts *and* goals. Thus, we propose an auxiliary alignment loss to regularize the dependencies of hypermodel $f_\phi(s, a, h_\psi(c_{\mathcal{T}}))$ and "*navigation computation*" Plan($\{s_i\}, c_{\mathcal{T}}, f_\phi$) on corresponding task context $c_{\mathcal{T}}$, to enable the hypermodel to be learned more accurately and effectively in the multi-task setting. In Figure 3 (right), we maximize the mutual information between task context $c_{\mathcal{T}}$ and *predicted* trajectories $\hat{\tau}_{\mathcal{T}}$ from the hypermodel on sampled tasks $\mathcal{T} \sim \rho(\mathcal{T})$:

$$\max_{h_\psi} \mathbb{E}_{\mathcal{T} \sim \rho(\mathcal{T})} \left[ I(c_{\mathcal{T}}; \hat{\tau}_{\mathcal{T}}) \right], \qquad (3)$$

where $h_\psi(c_{\mathcal{T}}) = \phi$ is the meta network predicting the weight of transition network $f_\phi$. Observe that: $I(\tau; c) = H(\tau) - H(\tau|c) \geq H(\tau) + \mathbb{E}_{\tau,c} \left[\log q(\tau|c)\right]$, we can equivalently optimize the RHS $\max_h \mathbb{E}_{\mathcal{T}} \left[\log q(\tau|c)\right] \iff \max_h \mathbb{E}_{(s,a,s')} \left[\log q(s'|s, a; h(c))\right]$, where we omit the subscripts for simplicity. This objective requires us to maximize the agreement of predicted states over the expectation of both (1) sampled states and (2) multiple tasks, i.e., minimizing distances $d(s', \hat{s'})$.

## 4 EXPERIMENTS

In the experiments, we assess our method and analyze its performance on DeepMind Lab (Beattie et al., 2016) maze navigation environment. We focus on the evaluation of zero-shot transfer in this section. We include more training results in Section B.1 of the appendix.

### 4.1 EXPERIMENTAL SETUP

We perform experiments on DeepMind Lab (Beattie et al., 2016), a reinforcement learning environment suite supporting customizing 2-D map layout. As shown in Figure 1, we generate a set of abstract 2-D maps, and use them to generate 3-D environments in DeepMind Lab. Each cell on the abstract map corresponds to 100 units in the agent world. In each generated map, all valid positions are reachable, i.e., there is only one connected component in the map. Given a sampled map, we then generate a start-goal position within a given distance range. Throughout each task, the agent receives the abstract map and start/goal location indicators, the joint state vector $o \in \mathbb{R}^{12}$ (consisting of position $\mathbb{R}^3$, orientation $\mathbb{R}^3$, translational and rotational velocity $\mathbb{R}^6$), and reward signal $r$. The action space is {forward, backward, strafe left, strafe right, look left, look right}, with an action repeat of 10. This means that, at maximum forward velocity, the agent can traverse a $100 \times 100$ block in two steps, but typically takes longer because the agent may slow down for rotations.

**Training settings.** We train a set of agents on a variety of training settings, which have several key options: (1) *Map size*. We mainly train on sets of $13 \times 13, 15 \times 15, 17 \times 17, 19 \times 19, 21 \times 21$ maps. One cell in the abstract map is equivalent to a $100 \times 100$ block in the agent's world. (2) *Goal distance*. During training, we generate start-goal pairs with distance between $1$ and $5$ in the abstract map. (3) *Map availability*. For each map size, we train all agents on the same set of 20 generated maps, with different randomly sampled start-goal pairs in each episode.

**Evaluation settings and metrics.** We have several settings for evaluation: (1) *Zero-shot transfer*. We mainly study this type of generalization, where the agent is presented with 20 unseen evaluation

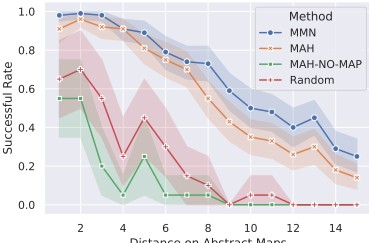 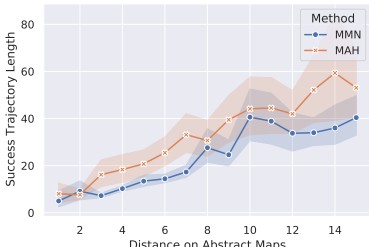

Figure 4: Zero-shot evaluation performance on $13 \times 13$ maps from distance 1 to 15. (a) Success rate of all agents. (b) Lengths of successful trajectories of MMN and MAH.

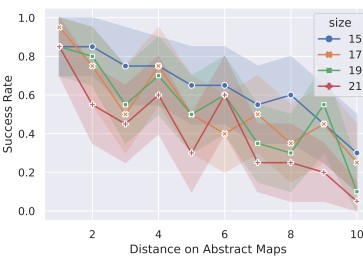 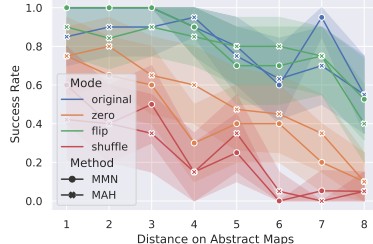

Figure 5: Study of performance on (a) different map sizes and (b) perturbation strategies.

maps, and has to navigate between randomly generated start-goal pairs of varying distances. (2) *Perturbation*. To understand how the map input affects the performance, we evaluate agents with input abstract maps perturbed by different strategies. (3) *Goal Distance* on abstract map. We focus on two scenarios of goal distance on **abstract maps**: *direct* navigation and *hierarchical* navigation. In the *direct* case, we evaluate on a range of distances ($[1, 15]$) on a set of maps, while in the *hierarchical* case, we generate a set of landmarks with a fixed distance of 5 between them and provide these to evaluated agents sequentially. (Metrics) We mainly report the average success rate over evaluated maps, and average length of successful trajectories, with $95\%$ confidence intervals.

**Methods.** We compare our model-based method, *Map-conditioned Multi-task Navigator* (**MMN**) with a task-conditioned model-free agent, *Map-conditioned Ape-X HER DQN* (**MAH**) and *Single-task Ape-X HER DQN* (**SAH**). MAH is based on *Ape-X DQN* and *HER*, which trains a *reactive* policy also conditioned on abstract 2-D map and goal input. SAH is a *single-task* model-free baseline, which does not have task context $c$ as input. We also use a **random** agent to demonstrate a lower bound of the navigation performance. Details of the MAH and SAH are in the appendix A.1.

### 4.2 ZERO-SHOT TRANSFER TO LOCAL GOALS IN NOVEL LAYOUTS

**Transfer of locally trained agents.** In this setting, we *train* all four agents on $20$ $13 \times 13$ maps with randomly generated local start-goal pairs with distance $[1, 5]$ in each episode. We train the agents until convergence; MAH typically takes $3 \times$ more training episodes and steps (see Section B.1). We *evaluate* all agents on $20$ *unseen* $13 \times 13$ maps and generate 5 start-goal pairs for each distance from 1 to 15 on each map. The results are shown in Figure 4. MMN and MAH generally outperforms the other two baselines. MMN has slightly better performance especially over longer distances, both in success rate and successful-trajectory length, even though it was only trained on distances $\leq 5$. Note that we did not include lengths for the two baseline methods due to their low success rates.

**Transfer on larger maps.** We also evaluated MMN on larger maps from $15 \times 15$ to $21 \times 21$. We tested the zero-shot transfer performance on 20 unseen maps of corresponding sizes and generated start-goal pairs with distance $[1, 15]$. As shown in Figure 5(a), even though training performance is similar among varying sizes (not shown), zero-shot transfer on novel larger maps becomes increasingly harder, which shows the difficulty of learning directly from abstract 2-D maps.

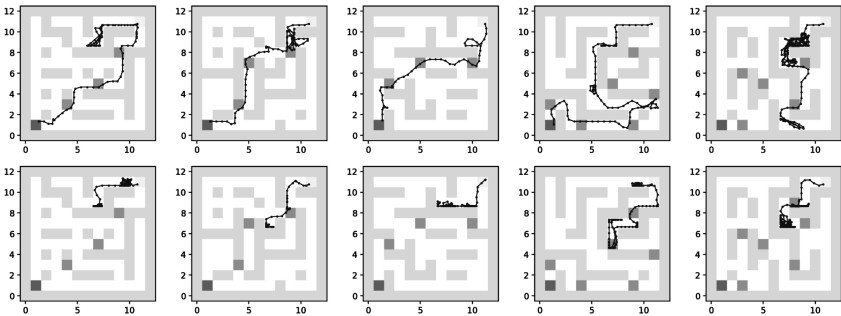

Figure 6: Trajectories from hierarchical navigation in zero-shot on $13 \times 13$ maps. The top-right corner is the start, and the bottom-left is the goal. Other darker cells are generated subgoals with distance 5. The first row is for MMN and second row is for MAH. For the first 4 tasks (columns), MMN successfully reached the goals, while MAH failed. Both methods failed in the last task.

|  | Success Rate | Average Success Length | Average Common Success Length |
|---|---|---|---|
| MMN | **16 / 20** | **62.25** | **46.44** |
| MAH | 9/20 | 92.55 | 92.55 |

Table 1: Zero-shot transfer results on hierarchical navigation

## 4.3 PERTURBATION OF ABSTRACT 2-D MAPS

To further study the importance of the abstract map input, during evaluation on $13 \times 13$ maps, we perturb the task context input $c = (m, g)$ and provide an disrupted version to agents. We examine several perturbation strategies: *zero*, *flip*, *shuffle*, and *original* accurate map input. (1) In *zero* mode, we give a tensor of zeros as abstract map and start/goal inputs to agents. In this case, the model may malfunction and rely on its policy and value prediction function to provide rough estimations. (2) In *shuffle* mode, we sample another map and start/goal pair within the set of evaluation maps. Thus, the entire structure should be entirely different and may largely affect the decision. (3) In *flip* mode, we randomly change the value of a cell with probability $p = 80\%$ (from wall to navigable space or vice versa). Note that flipping only changes the map $m$ but not the goal $g$, thus the agent can still capture a rough direction in local navigation. We evaluate on distances between 1 and 8.

As shown in Figure 5(b), *shuffle* has the largest effect to the performance, since randomly choosing another map not only changes the map input $m'$ to the agent, but also provides a misleading goal $g'$. Providing a *zero* map/goal has the second largest performance drop, since it does not mislead the agent with wrong map or goal, but does not provide the information that is necessary to complete the task without further learning and exploration. This demonstrates that both MMN and MAH are relying on the abstract map and start/goal input to do zero-shot navigation effectively. Surprisingly, the *flip* strategy turns out to have little performance decay. The reason for this may be that only the map is flipped, and since we evaluate on local start-goal pairs with distance $[1, 8]$, the flipping may not greatly affect the path connecting start-goal pairs, and the agent can rely on the unperturbed goal to navigate in the correct direction for short distances.

## 4.4 HIERARCHICAL NAVIGATION ON NOVEL LAYOUTS

In this section, we provide an additional *landmark oracle* to generate sequences of subgoals between long-distance start-goal pairs, and evaluate the performance of hierarchical navigation. We use the agent trained $13 \times 13$ maps, and evaluate on a set of 20 $13 \times 13$ unseen maps. On each map, we use the top-right corner as the global start position and the bottom-left corner as the global goal position, then plan a shortest path in the abstract 2-D map, and generate a sequence of subgoals with distance 5 between them; this typically results in 3 to 6 intermediate subgoals. Consecutive subgoal pairs are then provided sequentially to the agent as local start-goal pairs to navigate. The navigation is considered successful only if the agent reaches the global goal by the end.

We evaluate MMN and MAH on these 20 sequences of subgoals. We provide the next subgoal when the current one is *reached* or until *timeout*. As shown in Table 1, our model-based MMN outperforms the model-free counterpart by a large margin. MMN can reach 16 out of 20 global goals, which include all 9 successful cases of MAH. We also compute the average successful-trajectory length and average *common* success length, where the latter only considers the overlapping set of 9 tasks that both approaches succeeded in. We visualize five trajectories of zero-shot hierarchical navigation in Figure 6. The model-based MMN is more robust to the intermediate failed subgoals by navigating to the new subgoal directly, where the model-free MAH gets stuck frequently.

## 5 RELATED WORK

**Zero-shot navigation.** Navigation is widely studied in robotics, vision, RL, and beyond; to limit the scope, we focus on zero-shot navigation in novel environments, which is most relevant to this work. This excludes traditional approaches based on simultaneous localization and mapping (SLAM) (Thrun et al., 2005), since those methods need to explicitly build a map before navigation, and the map can only be used for the corresponding environment and cannot be transferred to other layouts. Learning-based methods such as by Mirowski et al. (2017; 2018) also require extensive training data from the same environment; they demonstrate generalization to new goals in the environment, but not transfer to new layouts. Jaderberg et al. (2017); Chen et al. (2019); Gupta et al. (2019); Chaplot et al. (2020) demonstrate agents that learn strategies to explore the new environment and potentially build maps of the environment during exploration; in contrast, we are interested in agents that do not need to explore the new environment. The approach of Gupta et al. (2019) does not necessarily explore the new environment; instead, it learns and exploits semantic cues from its rich visual input, which is orthogonal to our work since we use the state directly. Other domains such as first-person-shooting games have also used agents that navigate in novel environments (Lample & Chaplot, 2017; Dosovitskiy & Koltun, 2017; Zhong et al., 2020), but since navigation is not the primary task in those domains, the agents may not need to actually reach the specified goal (often none are specified). Most closely related to our work is by Brunner et al. (2018), who also use 2-D occupancy maps as additional input and perform experiments in DeepMind Lab. Their approach is very specific to map-based navigation, whereas our model-based approach uses more generic components such as hypermodels and MCTS, and may be more readily generalizable to other problems.

**Model-based RL and planning.** Model-based RL algorithms can be roughly grouped into four classes (Wang et al., 2019). 1. Train a transition model via supervised learning and learn on generated experience (Pong et al., 2018), or *Dyna-style*. 2. Follow the *analytical gradient* of the model (Heess et al., 2015), which may also require certain assumptions on model's form (Deisenroth & Rasmussen, 2011). 3. *Sampling-based planning* selects a promising next action by sampling from a certain action distribution (Hafner et al., 2018; Chua et al., 2018); this includes MCTS-based algorithms (Silver et al., 2017; Schrittwieser et al., 2019; Silver & Veness, 2010; Tian et al., 2019) in discrete-action setting. 4. Implicitly learn a model by matching value or reward predictions from unrolled trajectories to targets from real experience (Tamar et al., 2016; Oh et al., 2017; Schrittwieser et al., 2019). Additionally, a body of work (Parisotto & Salakhutdinov, 2017; Banino et al., 2018; Fortunato et al., 2019; Wayne et al., 2018; Ma et al., 2020) studies learning more structured latent models or representations useful for planning. Our method relies on types 3 and 4 above to learn an universal transition model on multi-task environments, and is most closely related to *MuZero* (Schrittwieser et al., 2019), which only predicts task-related quantities, such as rewards and values.

## 6 CONCLUSION

In this work, we have presented two approaches for enabling agents to navigate in environments with novel layouts without requiring further training or exploration (zero-shot), by using provided abstract 2-D maps and start/goal information. Both approaches MMN and MAH performed well in zero-shot navigation in short distances; for longer distances (with access to a landmark oracle), our model-based approach MAH performed significantly better. In future work, we will replace this oracle with a learned subgoal generator, extend this work to handle visual observation input and perform navigation in rich visual environments, and consider other types of provided task contexts.

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

# A    ADDITIONAL ALGORITHM DETAILS

## A.1    DETAILS ABOUT COMPARED METHODS

As a focus of our work, we examine the zero-shot transfer performance and compare all four agents in this section: MMN, MAH, SAH, and Random. In the implementation, we keep all components of MMN and MAH the same as much as possible, except that MAH only has a map-conditioned reactive policy network but no hypermodel. We use similar architecture for the Q-value network $Q(s, a, c_{\mathcal{T}})$, which is also conditioned on abstract 2-D maps and goals, as the policy and value prediction network $g_\theta(s, c_{\mathcal{T}})$ in our method. The task input to SAH has been further masked, so the Q-value network is simply a single-task version $Q(s, a)$. Thus, the main difference between our model-based approach MMN and the multi-task model-free variant MAH is that MAH entangles transfer on the map (dynamics) and goal (reward) levels, since the Q-value network needs to generalize value prediction jointly on different latent states $s$, goals $g$, and abstract 2-D maps $m$.

## A.2    ADDITIONAL DETAILS OF $n$-STEP RELABELLING

As shown in Figure 3 (right), we sample a trajectory of experience $(c_{\mathcal{T}}, \{s_t, a_t, r_t, s_{t+1}\}_t)$ on a specific map and goal $c_{\mathcal{T}} = (m_{\mathcal{T}}, g_{\mathcal{T}})$ from the replay buffer. Observe that, if the agent does not reach the goal area $\mathcal{S}_{\mathcal{G}}$ (a $100 \times 100$ cell in the agent space denoted by a coordinate $g_{\mathcal{T}}$ on the abstract 2-D map), it will only receive reward $r_t = -1$ during the entire episode until timeout. In large maps, this hinders the agent to learn effectively from the current map $m_{\mathcal{T}}$. Even if the agent partially understands a map, it would rarely experiences a specific goal area on the map again.[3] This is more frequent on larger maps in which possible navigable space is larger.

We adopt a multi-step strategy motivated by single-step HER, by relabelling failed goals to randomly sampled *future* states (visited area) from the trajectory. To relabel the task-conditioned bootstrapped $n$-step return, there are three steps. (1) *Goal.* Randomly select a reached state $s_t \in \mathbb{R}^{12}$ from the trajectories, then take the 2-D position $(x, y) \in \mathbb{R}^2$ in agent world and convert it to a 2-D goal support grid $g_{\mathcal{T}_S}$. Then, relabel the *goal* in task context $c_{\mathcal{T}_S} = (m_{\mathcal{T}}, g_{\mathcal{T}_S})$, keeping the abstracted map and start position unchanged. (2) *Reward.* Recompute the rewards along the n-step segment. In episodic case, we need to terminate the episode if the agent can reach the relabelled goal area $g_{\mathcal{T}_S}$, by marking "done" at the certain timestep or assigning zero discount after that step $\gamma_t = 0$ to mask the remaining segment. (3) *Value.* Finally, we need to recompute the bootstrapping task-conditioned value $v_n^{\mathcal{T}_S}, \pi_n^{\mathcal{T}_S} = g_\theta(s_t, c_{\mathcal{T}_S})$. Empirically, this strategy significantly increases the efficiency of our multi-task training by providing smoothing gradients when sampling a *mini-batch* of $n$-step targets from successful or failed tasks. It can also be applied to other multi-task agents based on $n$-step return.

## A.3    JOINTLY TRAINING HYPERMODELS

In the off-policy implementation, since we need to sample a mini-batch of trajectories, we have a *batch* of different contexts $[c_1, c_2, ..., c_n]$ during *multi-task* training, and need to generate a batch of weight $[\phi_1, \phi_2, ..., \phi_n]$ to compute each mini-batch gradient. To efficiently implement this, it is beneficial to use batch matrix multiplication in computing $[\phi_1, ..., \phi_n] = h_\psi([c_1, ..., c_n])$ and $[s'_1, ..., s'_n] = f([s_1, ..., s_n], [a_1, ..., a_n]; [s_1, \phi_2, ..., \phi_n])$ on a batch of randomly sampled transitions on different maps.

## A.4    ADDITIONAL DETAILS OF HYPERMODELS

We assume input is $M$-dimensional vectors in a mini-batch of size $B$, i.e., input tensor is $\mathbb{R}^{B \times M}$. The main network is a MLP with hidden units $L_i$ in layer $i$. We denote the input as layer 0, i.e., $L_i = M$. Thus, the weight matrix from layer $i - 1$ to layer $i$ has size $W_i \in \mathbb{R}^{L_{i-1} \times L_i}$. We assume the task context is $Z$-dimensional and in a mini-batch of input each has a different task context associated with it, i.e., input tensor to the hypermodel is in $\mathbb{R}^{B \times Z}$. The hyperwork is also a MLP with hidden units $K_i$ in each layer. The last hidden layer outputs a tensor $\mathbb{B} \times N$, where $N$ is the

---

[3]In our *extremely* low data regime, the agent only has one start-goal pair on a small set of map. While on *low* data regime, the agent can train on randomly sampled pairs on the maps. See the Setup for more details.

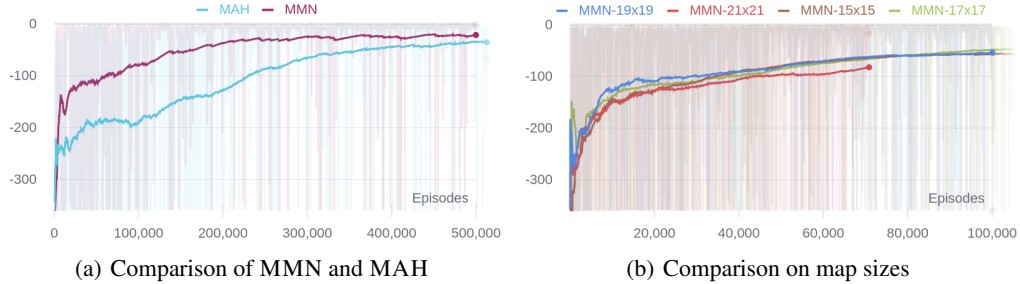

(a) Comparison of MMN and MAH       (b) Comparison on map sizes

Figure 7: Multi-task training performance of MMN and MAH.

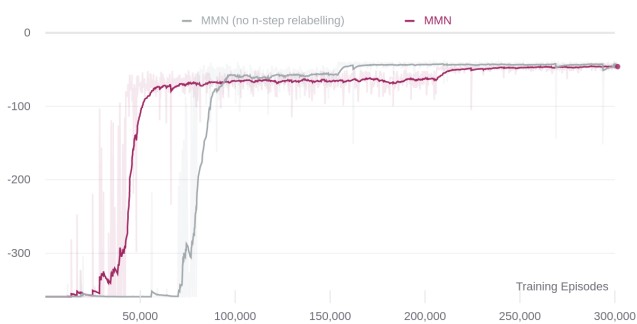

Figure 8: Ablation study of $n$-step relabelling.

dimension of output embedding for generating weights of the main network. The final generation layer has multiple branches. For generating weight tensor $W_i \in \mathbb{R}^{L_{i-1} \times L_i}$ of the main network *for all task contexts*, the branch outputs multi-task weight tensor $B \times L_{i-1} \times L_i$ and thus the mapping $h_{out} : B \times N \to B \times L_{i-1} \times L_i$ has weight tensor with dimensions $N \times (L_{i-1} \times L_i)$.

### A.5 ARCHITECTURE OF IMPLEMENTATION

Aiming at fair comparability of our method MAH with the model-free method MAH, we implement them in a unified framework. Both ours (model-based) and model-free baselines have $N$ actor workers, a single learner, and a centralized buffer worker. Each actor worker has a copy of the environment instance running single-threaded and take actions using either MCTS or a Q-network.

## B EXTRA EXPERIMENTAL RESULTS

### B.1 MULTI-TASK TRAINING PERFORMANCE

We demonstrate some representative results of the training performance of MMN and MAH. First, we compare the training on 20 of $13 \times 13$ maps with randomly generated goals at each episode, which is the most widely used training setting in our transfer evaluation. In Figure 7 (a), our model-based version MMN is much more sample efficient than the reactive MAH. There are two potential reasons: (1) model-based method is usually more sample efficient demonstrated in many single-task environments, and (2) our MMN is able to share knowledge between local patterns via the hypermodel. We also show the training results on larger map sizes with local start-goal pairs $[1, 5]$ in Figure 7(b). Although we found the evaluation performance decreases on larger maps, the local training performance w.r.t. episodes is similar. We include more results in the supplementary material.

We also study the ablation of $n$-step relabelling in Figure 8 in a special fixed goal setting on a $13 \times 13$ map. With the relabelling, our method is able to get signal earlier and learn faster.

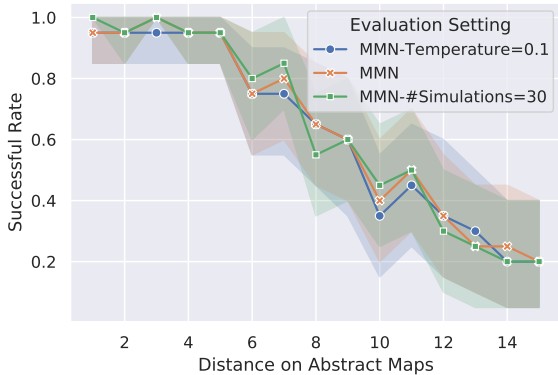

Figure 9: Ablation study of different hyperparameters in evaluation.

## B.2 ABLATION STUDIES OF TRANSFER

**Ablation study on evaluation hyperparameters.** We study two related hyperparameters in the zero-shot transfer: (1) number of *simulations*, where we change it to 30 from 100, and (2) *temperature* in action sampling, which is set to 0.1 from original value 0.25. We use one random goal for distance $[1, 15]$ on 20 of $13 \times 13$ maps. In Figure 9, we found decreasing number of simulations and increasing deterministicity do not produce significant difference.

**Transfer with few-shot adaptation.** To examine if zero-shot transfer still have any room of improvement, we also experiment on finetuning on same maps with different goals. Suprisingly, we found finetuning on novel maps does not result in improved performance. However, it is understandable since we have fully trained the agents during multi-task training, and the amount of data and learning steps in finetuning is insignificant compared to the training stage (about $10^5$ vs. $10^3$ steps, and $10^5$ vs. $10^2$ trajectories). We leave the further study of few-shot adaptation with abstract maps for future work.

