# OpenReview forum: "Model-based Navigation in Environments with Novel Layouts Using Abstract $2$-D Maps"
_ICLR.cc/2021/Conference — Reject_

### Official Review · AnonReviewer2 · 2020-10-21
**Interesting contribution to zero-shot navigation, falls a bit short on generalization**

**Rating:** 6
**Confidence:** 4

**Review:**

[EDIT AFTER DICUSSIONS] I thank the authors for their answer to my comments. I agree with the summary of the Area Chair and do not wish to modify my score.
[/EDIT]

##########################################################################
Summary:
This paper presents addresses the problem of zero-shot navigation in environments with novel layouts.  It introduces two approaches (MMN a model-based approach based on Monte-Carlo Tree Search, and MAH, a model-free approach based on Deep-Q networks). The paper also introduces n-step relabelling as a way to leverage failed trials and make learning more efficient. Experiments on the DeepMind Lab environment show that both methods perform well against a random baseline and that MAH extends better to larger maps.

##########################################################################
Reasons for score:
This paper presents a novel approach to an interesting problem. The method is sound and the approach rigorous. On the downside, an important claim of the paper is that the method is more generalizable than the latest work on this topic (Brunner et al. 2018) but this claim would have been stronger if supported with evidence, in particular with stronger baselines and more tasks.

 ##########################################################################
Pros:

- Clarity: the paper is well-structured, clear, and easy to read

- Impact: the paper addresses an interesting problem, in particular trying to use a general approach that is not specific to map-based navigation

- Rigor: the work presented in this paper is detailed and follows a clear methodology. Equations seem correct. The experimentation study is fairly detailed and the appendix provides significant details about the methods.

##########################################################################
Cons:

- No code is provided with the paper, making it hard for future work to use it as a baseline

- The paper mentions Brunner et al. 2018 as reference work. This work seems to achieve significantly better performance, at the "cost" of using a more task-specific, map-based navigation approach.This raises a few questions:
1. Could the authors have used Brunner et al. as a baseline for this work? Was the code available?
2. The authors (rightfully) claim that their approach is more generic and may be more readily generalizable to other problems. This statement would be a lot stronger if they actually proved it in the paper, i.e. if they used the same technic to solve a different problem.
3. The methodology used in Brunner et al. uses a larger variety of map sizes. The authors could have used this approach too to better evaluate their method.

- The paper provides a simple external baseline (random).  Could other (external) stronger baselines have been used, such as asynchronous advantage actor-critic ((Mnih et al.,2016) or model-free episodic control (Blundell et al., 2016)?

- The paper evaluates on a single task and could have been evaluated on more tasks to illustrate robustness to domain shifts, in particular in the light of the comment above regarding the generalization of the method. Examples include Jaco Arm, CoinRun, or the Surreal framework.

#########################################################################
Some typos:

- Page 2: "betweend" in the first paragraph
- Figures on page 6 are hardly legible on a printed version of the paper.  Try to make them larger maybe?
- missing upper cases in some references (e.g. POMDPs)

---

> ### Author Response · Authors · 2020-11-25
> **Response 2**
>
> We appreciate the reviewer for your valuable feedback and acknowledgment.
>
> Q = About Brunner et al. work, the map sizes
>
> We used the same map sizes as Brunner et al. (13x13 to 21x21).
>
> Brunner et al. use a more specific method to navigation, since they have a local direction prediction module given location, while we use hypermodel to understand maps. Also, they use a model-free agent, A2C, whose performance in standard RL benchmarks is worse than Ape-X. Ape-X [1] is the backbone of our MAH, a model-free baseline to the model-based MMN.
>
> Q = The baselines
>
> As we mentioned above, we used MAH as a strong model-free baseline to our model-based one, MMN. Compared to A2C and other model-free baslines, our MAH is based on Ape-X [1], a state-of-the-art off-policy model-free method. For MAH, we adopt it and also apply goal relabelling (HER).
>
> Q = Other environments
>
> We appreciate the suggestion and plan to work in other environments. However, it is potentially challenging, since we focus on environments that the necessary structure of corresponding MDPs can be given in order to solve the tasks in zero-shot.
>
> In our map-based navigation problem, this can be considered as a generalized version of goal-conditional RL. We need the agent to condition on a task context, which is a map and start-goal pair in our case, or task-conditional RL. This is far more difficult than goal-conditional RL, since optimal value function and policy mapping depend on both goal location and map layout $g(s, m, g)$.
>
> Our map-based planning is also different from the formulation of Meta-RL. Meta-RL usually assumes that the agent needs to learn a task context variable from the training experiences. In the adaptation phase, it needs to explore the environment to inference the posterior task context and then apply their task-conditioned reactive policy. We provide the agent with maps and goals as task contexts to achieve zero-shot navigation.
>
> [1] Horgan, Dan, et al. "Distributed prioritized experience replay." arXiv preprint arXiv:1803.00933 (2018).

---

### Official Review · AnonReviewer3 · 2020-10-28
**Concerns about task settings, lack of clarity in approach description**

**Rating:** 4
**Confidence:** 4

**Review:**

This paper tackles the task of going to a point-goal using an abstract 2D map of a given environment. The central idea is to use the given abstract 2D map to predict parameters for the transition function in the environment depicted by the map. This predicted transition function is used to search for actions to execute via planning.

Strengths: The paper tackles an interesting problem, that of how to use an abstract 2D map to navigate. It proposes interesting neural architectures for solving this task.

Shortcomings:
1. While the paper uses 3D environments from DeepMind lab, actual experiments in the paper employ a 12D agent state as input to the policies. Thus, in my view, the paper is actually only solving a 2D problem (even though Fig 1 shows a first-person image). If my understanding about the problem setup is correct, I wonder how would a hand-crafted transition function based on the map do -- we already know where is the free space, the starting location, the goal location, and the fixed scaling between the actual space and the map, can't we build a conservative transition model based on this information? How will such a hand-crafted solution compare? Conceptually, what is the utility of learning on top of such a hand-crafted transition function?

2. I found it challenging to understand the precise model. Section 3.2 is somewhat confusing: are we learning a policy \pi_t that mimics the outcome of a monte-carlo tree search? How are the functions $h_\psi$ and $f_\phi$ trained? What are $W_\phi$ in Figure 2? Without these details it is hard to fully understand the proposed model.

In summary, I am not convinced of the experimental setup being used to study the proposed approach. The proposed approach hasn't been described clearly enough to allow for a proper review.

Update: I thanks the authors for preparing a response and for providing additional experiments. I believe these additional experiments will benefit the paper. However, these experiments will require a full re-evaluation of the paper which, in my view, is beyond the scope of a response to a rebuttal.

---

> ### Author Response · Authors · 2020-11-25
> **Response 3**
>
> We thank the reviewer for your valuable feedback. We mainly detail the response to the main concern and answer some details.
>
> Q = The potential relation between path on abstract maps and agent environments
>
> We assume to not require planning on the given abstract maps (not plannable) and don’t need metric maps (the distance of points on maps does not necessarily correspond to agent environments). In other words, we treat maps as images and input to the hypermodel to parse with a convolution head. We did a set of experiments to break three components in the closed-loop map-based navigation: Map -- (1. planning) --> Path -- (2. action mapping) --> Environment -- (3. localizing) --> Map (repeat). In general, our learning-based agent is robust to these changes.
>
> For (1) planning, we try to break the implicit assumption of requiring perfect map topology. We adopt our hierarchical navigation setting, but generating subgoals on perturb maps. We show that path following (subgoals of distance 1) is less robust than local navigation (subgoals of distance 5) when maps and subgoals are inaccurate. However, our strategy can achieve higher performance even compare to path following. For (2) action mapping, we break the implicit requirement of known scaling between map and environment. We provide the agent with randomly transformed maps, where the ratio (in both x and y directions) is different. Our experiments showed that our agent does not rely on this knowledge or any perfect relation. For (3) localizing, we break the identifiability of agent position (a part of its joint state) by applying random noise. We aim to show that our agent does not rely on the position to understand the map, since providing position in the agent world has no relation with localizing on abstract maps and our learning-based method can adapt to the noise. The results confirmed no performance drop. In all these cases, one slight change could break a hand-crafted method, since it would require intensive perfect prior knowledge.
>
> Q = Why do we not want a path following problem?
>
> In more realistic settings, the input abstract map can be inaccurate, especially in terms of local connectivity. Therefore, we cannot expect direct planning on inaccurate maps to be reliable, since any local errors may result in dramatic changes of the behaviors for naive local path following. However, the agent can handle map errors since it navigates in the “true” environment. Also, intuitively, the navigation path itself can be more robust if we consider the “path following” problem in a higher map scale (longer goal distance), since there may exist more shared states between an optimal path and a path on the inaccurate map. This motivates us to adapt the agent behaviors to both map and (sub)goal and adopt a hierarchical strategy. We can sequentially assign subgoals to the agent, allowing it to navigate to the goals by understanding an (inaccurate) map by itself. By doing so, we want the agent to have planning ability on longer horizons and robustness to inaccurate map input.
>
> We empirically show this in breaking: Map -- (1. planning) --> Path. We found pure path following (subgoals of distance 1) on inaccurate maps can be even much worse than hierarchical navigation with learned local planners (subgoals of distance 5). In both cases, we use the flip strategy with a probability 20% and the planner between subgoals is our learned MMN. In conclusion, we expect the agent to be able to navigate to longer distances and thus more robust to local errors, instead of greedily following a path (simulated by subgoals of distance 1).
>
> Q = Some method details
> Yes, we learn a policy and value network predicting searched quantities as MuZero (Schrittwieser et al., 2019) does. The $W_\phi$ means weight matric of fully connected layers of the transition method, which is predicted by the meta-network $h_\psi$. The meta-network $h_\psi$ of the hypermodel is trained end-to-end by the gradients from task and model loss, while $f_\phi$ is predicted by $h_\psi$ and backpropagate its errors during training.

---

### Official Review · AnonReviewer1 · 2020-10-29
**Assumptions make the task very simplistic**

**Rating:** 4
**Confidence:** 4

**Review:**

This submission tackles the Point Goal navigation task given access to the agent’s starting location, current state (position and velocity), the goal location and a top-down map. The submission presents two approaches for tackling this task. First is MMN (Map-conditioned Multi-task Navigator) which is model-based approach which learns to hyper-network to convert map input to a transition function. Second approach is MAH (Map-conditioned Ape-X HER DQN) which is model-free approach using Ape-X DQN with Hindsight experience replay.

The submission has several weaknesses:
- I believe the key challenge in solving the point goal navigation task given access to a top-down 2-D map is localization. However, the authors assume access to the agent’s position and orientation which makes localization trivial. This assumption simplifies the task a lot. In fact, I believe there’s a deterministic solution to this task which would achieve ~100% success rate. The starting location, the goal location, and the map are given as input to the agent. The agent can use any planning algorithm like Dijkstra to plan the path to the goal location. The only minor challenge is that the action space is different from moving cells on the map, but the sequence of actions required to go to the next map cell is easy to obtain given that the actions and transitions are deterministic and the agent has access to its location and velocity at all time steps.
- There are several existing methods for the PointGoal navigation task in unseen environments without having access to the map (for example [1-4]), which seem to achieve much better performance (90-99% success) in more realistic environments. One or more of these methods should be used as baselines. A lot of these works have also open-sourced the code.
- The submission uses mazes in the DeepMind Lab for evaluation. I believe the key challenge of understanding abstract maps is highly simplified in this environment. I am not convinced that performance in this environment is indicative of performance in realistic environments as realistic maps are more complex and very different from mazes. There are several more realistic simulation environments available for navigation such as Habitat [5], Gibson [6], AI2Thor [7], etc.
- One of the desirables when learning to navigate with access to a 2-D map as opposed to navigating without maps is efficiency. Although the authors report success trajectory length, I believe Success weighted by path length (SPL) is a more informative metric to measure efficiency. SPL is commonly used in navigation papers in recent years (for example, [1-5]) and it makes it easier to compare the performance of different methods.

It is very difficult to access the efficacy of the method since the task is simplistic and there exists a deterministic solution to the task which would achieve ~100% success rate. The submission also lacks competitive baselines.

Suggestions for improvement:
- The agent should not have access to the true pose and velocity. This assumption makes the task trivial in my opinion.
- The actions and motion should not be deterministic.
- Empirical evaluation in a more realistic simulation environment would be more indicative of the efficacy of the proposed method.

[1] Ramakrishnan et al. Occupancy Anticipation for Efficient Exploration and Navigation. ECCV, 2020

[2] Wijmans et al. DDPPO. ICLR, 2020

[3] Chaplot et al. Learning to Explore using Active Neural SLAM. ICLR, 2020

[4] Sax et al. Learning to Navigate Using Mid-Level Visual Priors. CoRL, 2019

[5] Savva et al. Habitat: A Platform for Embodied AI Research. ICCV, 2019

[6] Xia et al. Gibson env: real-world perception for embodied agents. CVPR 2018

[7] Kolve et al. AI2-THOR: An Interactive 3D Environment for Visual AI. 2017

[8] Anderson et al. On Evaluation of Embodied Navigation Agents. 2018

---

> ### Author Response · Authors · 2020-11-25
> **Response 1**
>
> We thank the reviewer for your valuable feedback. We mainly detail the response to the main concern and summarize some potentially related work.
>
> Q = The potential relation between path on abstract maps and agent environments
>
> We assume to not require planning on the given abstract maps (not plannable) and don’t need metric maps (the distance of points on maps does not necessarily correspond to agent environments). In other words, we treat maps as images and input to the hypermodel to parse with a convolution head. We did a set of experiments to break three components in the closed-loop map-based navigation: Map -- (1. planning) --> Path -- (2. action mapping) --> Environment -- (3. localizing) --> Map (repeat). In general, our learning-based agent is robust to these changes.
>
> For (1) planning, we try to break the implicit assumption of requiring perfect map topology. We adopt our hierarchical navigation setting, but generating subgoals on perturb maps. We show that path following (subgoals of distance 1) is less robust than local navigation (subgoals of distance 5) when maps and subgoals are inaccurate. However, our strategy can achieve higher performance even compare to path following. For (2) action mapping, we break the implicit requirement of known scaling between map and environment. We provide the agent with randomly transformed maps, where the ratio (in both x and y directions) is different. Our experiments showed that our agent does not rely on this knowledge or any perfect relation. For (3) localizing, we break the identifiability of agent position (a part of its joint state) by applying random noise. We aim to show that our agent does not rely on the position to understand the map, since providing position in the agent world has no relation with localizing on abstract maps and our learning-based method can adapt to the noise. The results confirmed no performance drop. In all these cases, one slight change could break a hand-crafted method, since it would require intensive perfect prior knowledge.
>
> Q = Why do we not want a path following problem?
>
> In more realistic settings, the input abstract map can be inaccurate, especially in terms of local connectivity. Therefore, we cannot expect direct planning on inaccurate maps to be reliable, since any local errors may result in dramatic changes of the behaviors for naive local path following. However, the agent can handle map errors since it navigates in the “true” environment. Also, intuitively, the navigation path itself can be more robust if we consider the “path following” problem in a higher map scale (longer goal distance), since there may exist more shared states between an optimal path and a path on the inaccurate map. This motivates us to adapt the agent behaviors to both map and (sub)goal and adopt a hierarchical strategy. We can sequentially assign subgoals to the agent, allowing it to navigate to the goals by understanding an (inaccurate) map by itself. By doing so, we want the agent to have planning ability on longer horizons and robustness to inaccurate map input.
>
> We empirically show this in breaking: Map -- (1. planning) --> Path. We found pure path following (subgoals of distance 1) on inaccurate maps can be even much worse than hierarchical navigation with learned local planners (subgoals of distance 5). In both cases, we use the flip strategy with a probability 20% and the planner between subgoals is our learned MMN. In conclusion, we expect the agent to be able to navigate to longer distances and thus more robust to local errors, instead of greedily following a path (simulated by subgoals of distance 1).
>
> Q = Did we provide localization
>
> As we mentioned above, providing the position of the agent does not necessarily mean providing perfect localization on **map**. Also, we experimented on breaking the "Environment -- (3. localizing) --> Map (repeat)" part. Our results show that our learning-based agent is robust to this change.
>
> Q = How are [1-4] related?
>
> [2] does not use a map, so it needs to be explorative in novel environments, while we want the agent to be zero-shot (less explorative).
> [3] learns a map and local policy, while we provide an abstract map and learn a task-conditioned model-based planner. It can generalize its learned local policy and still require mapping in the test environment, but our planner is zero-shot generalizable given entirely different maps and corresponding environments.
> [1] is similar to [3] but focuses more on generalizing to different goals in environments with the same maps. SLAM-based methods may need more experience in test environments to build maps in prior.
> [4] generalizes to different visual scenes and is orthogonal to our work.
>
> In general, we use maze environments since its structure is topologically hard. We use joint states instead of images to (1) avoid semantic information in images and (2) focus on map level generalization, instead of visual features or other types.

---

### Official Review · AnonReviewer4 · 2020-10-29
**ICLR 2021**

**Rating:** 3
**Confidence:** 4

**Review:**


## Summary

 - The authors propose a method for "zero-shot navigation" that learns to navigate mazes from a map of the maze and the start and goal location in the maze.

## Strengths

 - This work builds on previous work in similar environment setups such as "Learning to Navigate in Complex Environments" (Mirowski et. al. 2017) to the "zero-shot" case, i.e., where the agent need not do any exploration when presented with a new environment but can immediately navigate to the goal.

## Weaknesses

 - My major concern about this work is that I just don't understand why this is a hard problem. You argue that "one cannot simply perform graph search on the 2-D map" but I fail to see why. Any classical planning method would have no trouble generating a plan for agent navigation from the 2-D map provided. The problem then, it seems to me, is reconciling the plan on the 2-D map with the actions that are needed to be taken in the actual environment. But given that there is a deterministic scaling between occupancy grid map and the agent world ("Each cell on the abstract map corresponds to 100 units in the agent world.") AND your agent's transition function is deterministic, then this problem is solvable zero-shot with a traditional planner (e.g. one based that generates a roadmap and then searches it but others would work)  and then executing the plan. Learning the dynamic model is also straightforward because you assume that you can directly observe the the joint state $o_t$.

 - Related to the above, you refer to the 2-D map as "abstract" but I fail to see what's abstract about it. It is a metric occupancy grid map that gives you full information about the layout of the environment.

 - As a result, my impression is that the solution is "over-engineered". There are many complex components and "hyper" models when something much simpler would have easily solved this problem.

## Minor Questions/Comments

 - At the onset you frame with this work with the classical SLAM literature, but this seems puzzling since you are providing the agent with both the map, and its initial location within the map, so both the mapping and localization are solved.

---

> ### Author Response · Authors · 2020-11-25
> **Response 4**
>
> We thank the reviewer for your valuable feedback. We mainly detail the response to the main concern. For the first part, we would like to claim that we **assume** the map is not plannable (simply like an image). We detail the reason for this in A2. We will answer the second part in Q1.
>
> Q1 = The potential relation between path on abstract maps and agent environments
>
> We assume to not require planning on the given abstract maps (not plannable) and don’t need metric maps (the distance of points on maps does not necessarily correspond to agent environments). In other words, we treat maps as images and input to the hypermodel to parse with a convolution head.
> We did a set of experiments to break three components in the closed-loop map-based navigation: Map -- (1. planning) --> Path -- (2. action mapping) --> Environment -- (3. localizing) --> Map (repeat). In general, our learning-based agent is robust to these changes.
>
> For (1) planning, we try to break the implicit assumption of requiring perfect map topology. We adopt our hierarchical navigation setting, but generating subgoals on perturb maps. We show that path following (subgoals of distance 1) is less robust than local navigation (subgoals of distance 5) when maps and subgoals are inaccurate. However, our strategy can achieve higher performance even compare to path following.
> For (2) action mapping, we break the implicit requirement of known scaling between map and environment. We provide the agent with randomly transformed maps, where the ratio (in both x and y directions) is different. Our experiments showed that our agent does not rely on this knowledge or any perfect relation.
> For (3) localizing, we break the identifiability of agent position (a part of its joint state) by applying random noise. We aim to show that our agent does not rely on the position to understand the map, since providing position in the agent world has no relation with localizing on abstract maps and our learning-based method can adapt to the noise. The results confirmed no performance drop. In all these cases, one slight change could break a hand-crafted method, since it would require intensive perfect prior knowledge.
>
> Q2 = Why do we not want a path following problem?
>
> In more realistic settings, the input abstract map can be inaccurate, especially in terms of local connectivity. Therefore, we cannot expect direct planning on inaccurate maps to be reliable, since any local errors may result in dramatic changes of the behaviors for naive local path following. However, the agent can handle map errors since it navigates in the “true” environment. Also, intuitively, the navigation path itself can be more robust if we consider the “path following” problem in a higher map scale (longer goal distance), since there may exist more shared states between an optimal path and a path on the inaccurate map. This motivates us to adapt the agent behaviors to both map and (sub)goal and adopt a hierarchical strategy. We can sequentially assign subgoals to the agent, allowing it to navigate to the goals by understanding an (inaccurate) map by itself. By doing so, we want the agent to have planning ability on longer horizons and robustness to inaccurate map input.
>
> We empirically show this in breaking: Map -- (1. planning) --> Path. We found pure path following (subgoals of distance 1) on inaccurate maps can be even much worse than hierarchical navigation with learned local planner (subgoals of distance 5). In both cases, we use the flip strategy with probability 20% and the planner between subgoals is our learned MMN. In conclusion, we expect the agent to be able to navigate to longer distances and thus more robust to local errors, instead of greedily following a path (simulated by subgoals of distance 1).

---

### Author Response · Authors · 2020-11-25
**Authors general response**

We thank all reviewers for their valuable feedback. Here, we summarize our responses and the additional experiments we did. We realized that there is a major concern about our problem settings and environments, so we will focus on this by outlining our responses. We will answer other concerns in individual responses.

We answer this major concern in several parts. (1) The potential relation between path on abstract maps and agent environments. (2) Why do we not want a path following problem? (Why do we assume the map is not plannable? Why don’t the “local planner” directly plan on maps?). (3) Also, we did a set of additional experiments breaking the potential relation to show that our method does not rely on them. The results are added to the Appendix. In general, we aim to break the implicit assumptions and prove the robustness of our learning-based method, where hand-crafted methods would be not trivial and not likely to succeed anymore.

Our additional experiments have three parts, corresponding to the three components in closed-loop map-based navigation: Map -- (1. planning) --> Path -- (2. action mapping) --> Environment -- (3. localizing) --> Map (repeat). We break each component and evaluate our agent. In general, our learning-based agent is robust to these changes, while one slight change could break a hand-crafted method, since it would require intensive perfect prior knowlege. We will detail them in individual responses.

---

### Decision · Program_Chairs · 2021-01-07
**Final Decision**

**Decision:**

Reject

**Comment:**

The paper considers the problem of 2D point-goal navigation in novel environments given access to an abstract occupancy grid map of the environment, together with knowledge of the agent's state and the goal location typical of point-goal navigation. The paper proposes learning a navigation policy in a model-based fashion, whereby the architecture predicts the parameters of the transition function and then uses this learned transition function to plan the agent's actions. The authors also describe a model-free approach that extends a version of DQN to reason over the 2D maps.

The paper was reviewed by four knowledgeable referees, who read the author response. The general problem of learning to navigate a priori unknown environments to reach a desired goal is an interesting problem that has received significant attention of-late in the learning community. In its current form, however, the paper does not adequately convey why this is a difficult problem that can not be solved using existing planning techniques or why it benefits from learning, particularly given access to an abstract map. These concerns apply more generally to point-goal navigation, namely the assumption that the pose of the agent and goal are fully known throughout (or the agent-relative pose of the goal) and that there is no uncertainty in the agent's motion. The practicality of these assumptions is unclear, and they are inconsistent with decades of research in robotics and robot learning, which addresses the more realistic setting in which there is uncertainty in pose and motion. The author response helps to clarify some of these questions, but it is still not fully clear why existing methods are insufficient for this task, whether they use traditional planning methods or are learned. Revisiting the discussion of why this is a hard problem would strengthen the paper, as would a more thorough evaluation that compares against other baselines.